# The Effect of a Modified Mindfulness-Based Stress Reduction (MBSR) Program on Symptoms of Stress and Depression and on Saliva Cortisol and Serum Creatine Kinase among Male Wrestlers

**DOI:** 10.3390/healthcare11111643

**Published:** 2023-06-03

**Authors:** Elham Mousavi, Dena Sadeghi-Bahmani, Habibolah Khazaie, Annette Beatrix Brühl, Zeno Stanga, Serge Brand

**Affiliations:** 1Department of Exercise Physiology, Payame Noor University (PNU), Tehran 19395-4697, Iran; 2Department of Psychology, Stanford University, Stanford, CA 94305, USA; bahmanid@stanford.edu; 3Department of Epidemiology & Population Health, Stanford University School of Medicine, Stanford, CA 94305, USA; 4Sleep Disorders Research Center, Kermanshah University of Medical Sciences (KUMS), Kermanshah 6715847141, Iran; 5Center for Affective, Stress and Sleep Disorders, Psychiatric Clinics of the University of Basel, 4002 Basel, Switzerland; annette.bruehl@upk.ch; 6Centre of Competence for Military and Disaster Medicine, Swiss Armed Forces, 3008 Bern, Switzerland; 7Division of Sport Science and Psychosocial Health, Department of Sport, Exercise and Health, University of Basel, 4052 Basel, Switzerland; 8Substance Use Prevention Research Center, Kermanshah University of Medical Sciences (KUMS), Kermanshah 6715847141, Iran; 9School of Medicine, Tehran University of Medical Sciences (TUMS), Tehran 1416634793, Iran; 10Center for Disaster Psychiatry and Disaster Psychology, Psychiatric Clinics of the University of Basel, 4002 Basel, Switzerland

**Keywords:** mental health, mindfulness-based stress reduction (MBSR), cortisol, athletes, creatine kinase

## Abstract

Objectives: The aims of the present study were two-fold: to investigate whether, compared to an active control condition, a modified mindfulness-based stress reduction (MBSR) program could (1) reduce symptoms of stress and depression, and (2) regulate salivary cortisol and serum creatine kinase (CK) concentrations, two physiological stress markers. Methods: Thirty male wrestlers (*M*_age_ = 26.73 years) were randomly assigned either to the MBSR intervention or the active control condition. Both at the beginning and at the end of the intervention, the participants completed questionnaires on perceived stress and depression; in parallel, salivary samples were collected to measure cortisol in saliva, while blood samples were collected to assess serum CK. The study lasted for eight consecutive weeks. The intervention consisted of 16 group sessions (90 min each); the active control condition had an identical schedule, though without bona fide interventions. During the study period, the participants kept their sleeping, nutritional and exercising schedules unaltered. Results: Over time, symptoms of stress and depression decreased; the level of decrease was more prominent in the MBSR condition than the active control condition (significant p values and large effect sizes of interaction). Further, cortisol and creatine kinase concentrations also decreased more in the MBSR condition compared to the active control condition (large effect sizes of interaction). Conclusions: The present study’s findings suggest that among male wrestlers, a modified MBSR intervention have the potential to reduce both psychological (stress and depression) and physiological (cortisol and creatine kinase) indices as compared to an active control condition.

## 1. Introduction

Wrestling is characterized by short periods of both low-intensity and high-intensity activities. Frequently competing in a challenging environment not only requires different physiological preparation, but there is also a high demand for psychological aspects, such as coping with stress [1]. Placing a high value on these psychological variables are critical for achieving the best results at the highest levels of competition [2].

A competitive context is generally regarded as both a psychological and physiological stressor [3,4]; in some studies, higher cortisol concentrations mirroring physiological stress were observed in individuals in competitive environments [3,4]. In addition to symptoms of stress, elite athletes may experience some degree of depression, while often they are not aware of this psychological health issue. However, symptoms of depression may negatively impact an athlete’s performance [5,6]. Kumartasli et al. [7] showed that wrestlers reported higher scores for depression than taekwondo athletes. Therefore, stress and depression are two psychological variables that deserve more attention [8].

While various methods, such as cognitive behavioral therapy [9], reduce symptoms of depression and stress, in the present study we applied the mindfulness-based stress reduction (MBSR) program, as MBSR has proven to be a valuable intervention technique in improving a broad range of psychological and physiological health indicators [10].

Mindfulness is defined as a cognitive process without a judgmental attitude and with the awareness of the present moment [11,12,13]. Among the variety of mindfulness programs, mindfulness-based stress reduction (MBSR) is well known to diminish the psychobiological effects of stress response and depression among different populations [11]. Typically, MBSR demands 2.5 to 3 h per session and per week scheduled for a total of eight weeks. The program consists of body scan, behavioral and emotional awareness, yoga, breathing techniques, meditation practices, awareness of thoughts, and focusing on the current moment [11] (see also Table 1). Bühlmayer et al. [14] showed that among athletes, mindfulness interventions had a favorable impact on performance-related psychophysiological parameters, such as salivary cortisol and anxiety, as compared to (passive) control conditions [14]. In another study, a mindfulness-based intervention improved self-compassion and grit among young adult female elite athletes when compared to an active control condition [15]. While mindfulness has the power to alleviate negative thoughts and to promote mental and physical health [16], a meta-analysis [14] showed that it also had a favorable impact on physiological processes, including resting heart rate, and dimensions of the immune system among athletes in various fields as compared to (passive) control conditions [14]. Indeed, the effectiveness of mindfulness-based strategies on physiological variables has been shown in previous studies [17]. It has been proven that high levels of cortisol were associated with complex mental health issues, such as symptoms of depression and stress [18,19]. MacDonalds and Minahan [17] observed that salivary cortisol diminished after a standardized eight-week mindfulness program in wheelchair basketball players [17]. Further, among soccer players, an 6 min mindfulness-based intervention reduced salivary cortisol concentrations [20].

In addition to cortisol, creatine kinase is another biomarker used to indicate the physiological stress level and to measure the degree of tissue damage [21]. After acute exercise, CK concentrations increased following muscle injuries [22] and CK levels increased after low to moderate training intensities following mild muscle tissue damage [23]. After 12 sessions of weight lifting, serum CK concentrations increased by about 29% [24]. Erdogan [25] demonstrated that 12 weeks of regular volleyball training at a maximal heart rate raised the level of muscle injury biomarkers, such as CK and CK-MB (myocardial izoenzyme) [25]. Most importantly, a recent study reported a higher level of total CK and CK-MB in individuals with depressive disorders [26]. Given this background outlined above and for lack of previous results, we explored, whether and to what extent a modified MBSR intervention might reduce CK concentrations.

Taking the aforementioned background details into account, including the facts that wrestlers appeared to report particularly more stress levels compared to taekwondo athletes [27] and that no studies has been performed so far on male wrestlers, the aims of the present study were to examine the impact of a modified MBSR program on psychological dimensions, namely stress and depression, and physiological dimensions, namely salivary cortisol and serum CK, compared to an active control condition.

Based on previous results [14,16,17], we expected that both psychological and physiological stress indices would decrease over time in the MBSR condition compared to the active control condition.

## 2. Materials and Methods

### 2.1. Participants

Thirty male wrestlers (mean age: 26.73 years (SD = 0.80); mean height = 174.83 cm (SD = 1.59); and mean body weight = 77.36 kg (SD = 15.20)) participated in the present study. Participants were fully informed about the aims of this study and about the secure and anonymous data handling. Thereafter, participants signed the written informed consent. Inclusion criteria included the following: 1. Age 18 years or higher; 2. Male gender at birth; 3. Active wrestler for about 8 years and exercising at least 4.5 h/week; 4. Willing and able to comply with the study conditions, as outlined in more details below; and 5. Signed written informed consent. Exclusion criteria were as follows: 1. Injury during the study period; and 2. Experienced in or currently attending MBSR programs or very similar courses, such as progressive muscle relaxation, meditation, yoga or autogenic training. The Ethical Committee of the Kermanshah University of Medical Sciences (KUMS; Kermanshah, Iran; register code: IR.KUMS.REC.1401.247) approved the study, which was performed in accordance with the seventh and current revision [28] of the Declaration of Helsinki.

Of the 43 eligible athletes assessed, 30 (69,76%) were included in the study at baseline. The thirteen athelets were dropped because they did not meet the inclusion criteria: eight atheletes reported to be experienced in MBSR techniques or to currently attending courses on relaxations techniques, two were injured between the assessment and baseline and three did not provide the written informed consent. Further, it is to be noted that once a participant was assigned to either the MBSR or active control condition, there were no drop-outs.

### 2.2. Sample Size Calculations

To calculate the sample sizes using G*Power [29], we referred to a previous study [15], and used the following indices: effect size f: 0.55; alpha: 0.05; power: 0.95; number of groups: 2; number of measurements: 2; and total sample size: 24. To counterbalance possible dropouts, it was decided to include at least 30 participants.

### 2.3. Procedure

To avoid possible psychological and physiological alterations, this whole study was performed during the off-season of competition. Next, to comply with the study conditions, participants were asked to continue their regular life style as follows: 1. Continuing their regular exercising of three sessions in a week for about 80–90 min/session; 2. Keeping their sleeping schedules stable, with 7 to 8 h sleep/night both during the weekdays (Saturday to Wednesday) and weekend days (Thursday and Friday); 3. Keeping a well-balanced diet schedule of three meals/day consisting of food with low lipids, low carbohydrates, higher protein and fibers, including regular liquid intake (including 2–3 cups of tea/day) and 4. Abstaining from mood- and arousal-altering substances, such as excessive coffee, energy drinks, medications, or illegit substances). At baseline, all the participants completed a booklet of questionnaires covering sociodemographic information and symptoms of stress and depression, and they provided saliva and blood samples under standardized laboratory conditions (see below). An identical assessment was performed eight weeks later at the end of the study. Further, at baseline, the participants were randomly assigned either to the MBSR or to the active control condition.

### 2.4. Measures

#### 2.4.1. Psychological Dimensions

##### Perceived Stress

To self-assess stress, participants completed the Farsi version [30] of the Perceived Stress Scale [31]. It consists of 14 items, and answers are given on 5-point Likert scales; sum scores range from 0 to 40, with higher scores reflecting a more pronounced level of stress (Cronbach’s *α* = 0.86).

##### Symptoms of Depression

To self-assess symptoms of depression, participants completed the Farsi version [32] of the Beck Depression Inventory. The questionnaire consists of 21 items related to symptoms of depression [33]. Answers are given on 4-point Liker scales; scores ranged from 0 to 63, with higher scores indicating more substantial depression (Cronbach’s *α* = 0.82).

#### 2.4.2. Physiological Dimensions

##### Salivary Cortisol

To assess cortisol concentrations in saliva, we strictly followed the algorithm suggested in [34] (see also [35,36,37,38]). Four saliva samples were taken; participants provided saliva samples in the so-called “Salivette”, a proven, quick and hygienic device for saliva sampling (Sarstedt, Nümbrecht, Germany). The first sample was taken immediately after waking up in the morning at about 6.30 am, followed by three saliva samples collected in 15 min intervals. During sampling, participants were asked to fast and not brush their teeth to avoid micro-injuries of the tissue. Participants brought their saliva samples to the lab within the following 1 h. Samples were centrifuged at 4 °C (2000 rpm, 10 min) and stored at −20 °C until assay. Next, free salivary cortisol concentrations were analyzed using a time-resolved immunoassay with fluorometric detection ‘‘Coat-A-Count’’ Cortisol RIA from DPC (Diagnostics Products Corporation; obtained through H. Biermann GmbH, Bad Nauheim, Germany) as described in detail elsewhere [39,40]. Intra- and inter-assay variability was less than 3.5% and 4.20%, respectively.

##### Creatine Kinase

For measuring the serum level of total CK, blood samples were taken in the lab on the first and the last morning of the study after two days without exercising and keeping their individual standard diet. Participants arrived in the lab at about 8 am, always in a fasting state. The resting state lasted 15 min; thereafter, 3 ml venous blood was extracted. All samples were centrifuged at 3000 rpm for 10 min and kept at −80 °C until analysis. To analyze creatine kinase, we followed a protocol previously decsribed [41,42]. The activity of creatine kinase was determined via the spectrophotometric technique coupling creatine kinase with pyruvate kinase and lactate dehydrogenase followed at 340 nm. Additionally, the rate of reaction was measured following the oxidation of NADH.
$$creatine+ATP\;\overset{CPK}{\to }\;creatine\;phosphate+ADP$$
$$ADP+phosphoenopyruvate\;\overset{Pyruvate\;kinase\;}{\to }\;ATP+pyruvate$$
$$pyruvate+NADH+H^{+}\;\overset{Lactate\;dehydrogenase}{\to }\;lactate+NAD^{+}$$

#### 2.4.3. Interventions

##### MBSR Intervention

Mindfulness training was performed as group interventions at the Sleep Disorders Center of Kermanshah University of Medical Sciences (Kermanshah, Iran) twice a week, for eight consecutive weeks. Each session lasted for about 120 min. Instructors were trained and certified in MBSR and they had extensive experiences in conducting MBSR programs.

Table 1 depicts the modified MBSR intervention.

##### Active Control Condition

Participants in the control group condition met twice a week at the School of the Health University of Kermanshah (Kermanshah, Iran). They were asked questions about their physical and psychological conditions. Exercises and general life situations were discussed. They also had group discussions on the topics mentioned in recent newspapers and magazines. The primary purpose of this gathering was to eliminate or minimize the effect of social contact and group friendship in the experimental groups. It is to be noted that the active control condition was not a *bona fide* psychotherapeutic intervention to improve cognitive–emotional processes [43].

#### 2.4.4. Analytical Plan

Mean and standard deviations were used to report the values of the measured variable (mean ± SD). The normality of variables was assessed using the Shapiro–Wilk test, and the homogeneity of variance was assessed using the Levene’s test. Four ANOVAs for repeated measures were performed with the following factors: time (baseline; study end); group (MBSR; control condition), and time by group interaction. Dependent variables included symptoms of stress and depression, saliva cortisol concentrations and blood CK concentrations. The level of significance of *p* < 0.05 was considered for all calculations. For F-tests, we followed the statistical indices reported by Cohen [44,45]. Effect sizes were reported as partial eta-squared [η_p_^2^]) and interpreted as follows: trivial (T) = 0.019 < η_p_^2^; small (S) = 0.020 ≤ η_p_^2^ ≤ 0.059; medium (M) = 0.06 ≤ η_p_^2^ ≤ 0.139; or large (L) = η_p_^2^ ≥ 0.14. Statistical analyses were conducted using SPSS^®^ version 28.0 (IBM Corporation, Armonk, NY, USA) for Apple^®^.

## 3. Result

Table 2 reports the descriptive statistical indices of symptoms of stress and depression, and cortisol and CK concentrations for the MBSR and the active control condition, both at the baseline and at the end of the study. Table 3 reports the inferential statistical indices (ANOVAs for repeated measures with the factors time, group, and time by group interaction).

### 3.1. Stress

Symptoms of stress decreased over time (*F*(1,28) = 72.47, *p* < 0.05 η_p_^2^ = 0.72 (L), Wilk’s λ= 0.28); the level of decrease was more in the MBSR condition than the control condition (*F*(1,28) = 47.60, *p* < 0.05, η_p_^2^ = 0.63 (L)). No statistically significant mean differences were found between the two groups (*F*(1,28) =1.07, 0.04 (S)).

### 3.2. Depression

Symptoms of depression decreased over time (*F*(1,28) = 53.28, *p* < 0.05 η_p_^2^ = 0.66 (L Wilk’s λ= 0.66)); the level of decerease was more in the MBSR condition compared to the active control condition (*F*(1,28) = 1.79, *p* < 0.05, η_p_^2^ = 0.66 (L). Moreover, compared to the active control condition, participants in the MBSR condition reported lower scores for depression (*F*(1,28) =6.79, η_p_^2^ =0.20 (L)).

### 3.3. Saliva Cortisol

Levels of salivary cortisol decreased over time; the level of decerease was more in the MBSR group (*F*(1,28) = 26.92, *p* < 0.05 η_p_^2^ = 0.49 (L), Wilk’s λ= 0.51) compared to the control group (*F*(1,28) = 19.57, *p* < 0.05, η_p_^2^ = 0.41 (L)). No statistically significant mean differences were found between the two groups (*F*(1,28) =0.35, η_p_^2^ = 0.03 (S)) (see also Figure 1).

### 3.4. Serum Creatine Kinase

Over time, the serum creatine kinase levels decreased; the reduction was more prominent in the MBSR group (*F*(1,28) = 32.25, *p* < 0.05, η_p_^2^ = 0.54 (L), Wilk’s λ = 0.47) compared to the control group (*F*(1,28) = 19.45, *p* < 0.05, η_p_^2^ = 0.41 (L)). The analysis indicated that there was a significant mean difference between the two groups (*F*(1,28) = 6.44, η_p_^2^ = 0.19 (L)) (see Table 2 and Table 3).

## 4. Discussion

The aims of the current study were to determine whether an eight-week modified MBSR program had a favorable impact on the psychological dimensions, namely stress and depression, and physiological dimensions, namely saliva cortisol and serum creatine kinase (CK), of male wrestlers compared to an active control condition group. Results showed that MSBR had a favourable impact. Over the course of the program, symptoms of stress and depression, and saliva cortisol and serum CK decreased; the level of decrease was more prominent in the MBSR condition compared to the active control condition. The results obtained in this study add to the current literature in the following important ways: 1. A modified MBSR program improved both psychological and physiological stress parameters; 2. Unlike the majority of previous studies, our MBSR program was compared to an active control condition; in doing so, the impact of a possible favorable social influence on outcome variables could be controlled; 3. To our knowledge, this was the first approach to investigate the impact of a modified MBSR program on both psychological and physiological dimensions among male wrestlers (for adult female athletes, see [15]); and 4. The focus on wrestlers appears to be particularly important, as wrestlers appeared to be at a higher risk for symptoms of depression compared to athletes of other sport disciplines [7].

Two psychological indicators (perceived stress and depression) have been reported in the present study concerning MBSR training. The psychological benefits of MBSR have been proven in a wide variety of sports. Generally, our findings expand on the previous studies researching the effectiveness of MBSR on psychological and physiological parameters of athletes. To illustrate a previous study, perceived stress was examined in participants with different sports background, and Di Fronso et al. [46] showed that 10 sessions of MBSR reduced stress in both recreational and professional athletes [46]. Psychological components, such as stress and depression, decreased after nine weeks of MBSR in young female rowers compared to a passive control condition [47]. Norouzi et al. [48] analyzed the impact of 16 sessions of MBSR on Iranian retired football players and showed that the MBSR strategy promoted better psychological wellbeing by reducing stress and depression levels [48]. Previously, MBSR interventions on other aspects of wrestlers’ psychological wellbeing, namely anxiety, self-confidence, and mental toughness, were analyzed [49]; nevertheless, the present study was the first to measure depression and stress after MBSR training in male wrestlers.

Regarding the high level of cortisol and stress levels pre-and post- competition in different wrestling styles, it is suggested that psychological interventions should be employed to decrease the psychophysiological response to stress [3,4]. According to Ghahremani et al. [50], the method used for stress reduction should have an impact on salivary cortisol concentration and perceived stress in wrestlers; as shown in the recent study, MBSR has led to a significant reduction in both variables [50]. Along similar lines, MacDonald and Minahan [17] have investigated the effect of an eight-week standardized MBSR program on alleviating the increase in salivary cortisol during the competition period in elite wheelchair basketball players. They observed that in the control group, elevated cortisol concentration decreased after the sixth week; however, in the MBSR group, the reduction occurred already after two weeks. Therefore, MacDonald and Minahan [17] demonstrated that mindfulness training could be a practical method to manage elevation in stress-induced cortisol levels [17].

Despite the fact that modifications in serum CK stimulated by strenuous exercise in response to either muscle injury or hypertrophy is a natural physiological process, during recovery time, the athletes with lower levels of serum CK is at an advantage, which might be due to their resistance to the muscle damage process [51]. As participants in the present study continued their regular life style, including training sessions, sleep, nutritional schedules, and leisure time activities, throughout the period of study and used MBSR as a resuming session, it can be presumed that the MBSR protocol accelerated recovery time and muscle damage recovery [52]. Various meta-analyses highlighted that lowered CK levels are a hallmark of more efficient post-workout recovery outcomes obtained with the implementation of a variety of methods and therapies [53]. However, unlike the present study, no study has directly considered the impact of MBSR on serum CK during recovery time in wrestling.

Our study design does not allow to gain a deeper understanding of the underlying cognitive–emotional processes to explain why the administration of MBSR protocol improved both psychological and neuroendocrine indices. Thus, we can only speculate about the factors to which the observed changes in stress, depression, cortisol and CK can be attributed to. As described in Table 1, the MBSR intervention program entails perceiving cognitions and emotions without reacting to them, accepting one’s personality traits, remaining positively disposed and experiencing moments of time in peace. More specifically, as outlined in [12,15,47,54,55], MBSR does not aim to change the thought contents, emotions or bodily sensations, but rather aims to focus on how one relates to such experiences. In line with these observations, researchers [12,55,56,57] argued that mindfulness-based programs bring about mindfulness via an attitude of awareness, acceptance, kindness, openness, patience, non-striving, equanimity, curiosity and non-evaluation. Given this, we claim that such cognitive–emotional processes also occurred among the participants in the MBSR condition, leading to both improved psychological and physiological changes.

Despite the novelty of the results obtained in this study, several limitations warrant against the overgeneralization these results. First, the sample size was small; though, statistical calculations also relied on effect sizes, which do not vary as a function of sample size. Second, it is possible that latent and unassessed factors, such as higher scores for anxiety, insecurity and low self-esteem, along with unknown individual exercise schedules, might have biased two or more variables in the same or opposite directions and that, accordingly, the present pattern of results might be biased. Moreover, we investigated self-assessed symptoms of depression, while such self-ratings do not reflect a diagnostic and expert-rated psychiatric category. Third, latent and individual differences of CK concentrations might have biased the present results; more specifically, CK concentrations may depend on age, gender, race, muscle mass, physical activity and climatic condition [58]. As such, it is probable that the reported means and standard deviations may have obscured individual values, as the standard deviations (Table 2) appear to suggest. As can be seen, means and standard deviations of baseline CK concentrations appeared unbalanced between the groups, though the effect size was spurious (Cohen’s d = 0.02). More specifically, while participants were asked to keep their regular lifestyle stable, including exercise, nutrition and sleeping schedules, and leisure time activities, so as to keep the CK concentrations as standardized as possible (see [58]), futures studies should pay particular attention to standardizing the CK assessments. Fourth, the study design did not include an estimate of the long-term effects of MBSR on psychological dimensions, namely stress and symptoms of depression, and on physiological parameters, namely saliva cortisol and serum CK concentrations. Fifth, only male wrestlers were assessed; accordingly, it remains unclear whether and to what extent the present results might be also observed among female wrestlers.

## 5. Conclusions

Compared to an active control condition, an eight-week modified MBSR intervention improved both psychological parameters, namely symptoms of stress and depression, and physioological parameters, namely morning salivary cortisol and serum CK, among young adult male wrestlers. The present data corroborated and expanded on previous research in the field of MBSR, sports psychology and sports physiology.

## Figures and Tables

**Figure 1 healthcare-11-01643-f001:**
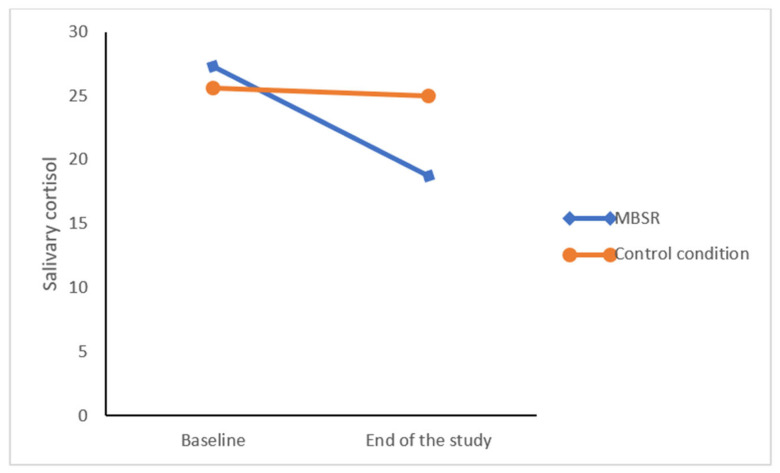
Change in saliva cortisol indices over time within and between the MSBR and the control condition. Points represent the means.

**Table 1 healthcare-11-01643-t001:** Overview and detailed description of the MBSR intervention.

Week 1 (The training stage)	- Introducing MBSR - Representing the sessions schedules - Forming groups - Expressing external experiences of depression
Week 2 (The stage of communication and conceptualization of mindfulness)	- Practicing to pay attention to the present moment without judgements - Practicing being aware of inner experiences - Practicing mindfulness of breathing, Yoga - Teaching behavioral flexibility in coping with depression
Week 3 (The stage of practicing and encountering with the mind)	- Reviewing assignments of previous sessions - Teaching to avoid being dependent on failures and negative thoughts - Body scan - Muscle relaxation - Teaching breathing techniques - Attentional Technique Training (ATT).
Week 4 (The stage of identifying values, and distinguishing between value and purpose)	- Teaching to look at their goals and values without negative or positive judgments. - Reviewing last week’s homework - Seated meditation - Three-minute breathing technique
Week 5 (The stage of practicing mindfulness and staying in the present)	- A five-minute “see or hear” exercise - Practicing breathing mindfulness - Performing yoga, muscle relaxation, meditation, and body scan
Week 6 (The cognitive flexibility, emotional and behavioral stage)	- Teaching not to assume that feelings, emotions, and thoughts of depression are irreversible - Practicing judgment and acceptance - Breathing techniques and sitting meditation - Explaining stress and its relationship with depression, thoughts, and physical sensations
Week 7 (The stage of acceptance)	- Teaching to be aware of personal reactions - Accepting negative thoughts - Conscious Yoga, sitting meditation, body awareness
Week 8 (The stage of combination and cohesion)	- Sleep hygiene - Repeating practices from previous sessions - Making a list of enjoyable activities - Applying MBSR in daily life

**Table 2 healthcare-11-01643-t002:** Descriptive statistical indices of symptoms of stress and depression, and saliva cortisol concentrations and serum creatine kinase concentrations at the baseline and at the end of the study, for the MBSR and the active control condition.

	MBSR Condition		Active Control Condition	
	Baseline	Study end	Baseline	Study end
	M (SD)	M (SD)	M (SD)	M (SD)
Stress	19.43 (4.09)	10.45 (2.97)	16.76 (4.40)	15.82 (4.00)
Depression	14.10 (2.15)	9.97 (1.94)	15.61 (2.77)	12.76 (3.34)
Cortisol (nmol/L)	27.27 (7.01)	18.70 (5.31)	25.64 (7.88)	24.96 (7.65)
Creatine kinase (U/L)	430.40 (236.60)	327.53 (186.02)	568.13 (184.58)	555.20 (184.57)

Data are reported as mean and standard deviation.

**Table 3 healthcare-11-01643-t003:** Inferential statistical indices related to psychological wellbeing (stress and depression), and the levels of serum creatine kinase and salivary cortisol.

	Time	Group	Time × Group Interaction	Wilks’ Lambda
	F η_p_^2^	F η_p_^2^	F η_p_^2^	
Stress	*F*(1,28) = 72.47 * 0.72 (L)	*F*(1,28) = 1.07 0.04 (S)	*F*(1,28) = 47.60 * 0.63 (L)	0.28
Depression	*F*(1,28) = 53.28 * 0.66 (L)	*F*(1,28) = 6.79 * 0.20 (L)	*F*(1,28) = 1.79 * 0.656 (L)	0.66
Cortisol (nmol/L)	*F*(1,28) = 26.92 * 0.49 (L)	*F*(1,28) = 0.35 0.03 (S)	*F*(1,28) = 19.57 * 0.411 (L)	0.51
Creatine kinase (U/L)	*F*(1,28) = 32.25 * 0.54 (L)	*F*(1,28) = 6.44 * 0.19 (L)	*F*(1,28) = 19.45 * 0.41 (L)	0.47

Note: η_p_^2^ = partial eta squared, * = *p* < 0.5, S = small effect size, M = medium effect size, and L = large effect size.

## Data Availability

Data are made available to explicit experts in the field. Such experts should clearly formulate their hypotheses; further, they should fully describe, how and where the do securely store the data file, and how they make sure that the data file is not shared with and securely protected from third parties.

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
