# Peer review of "The Effect of a Modified Mindfulness-Based Stress Reduction (MBSR) Program on Symptoms of Stress and Depression and on Saliva Cortisol and Serum Creatine Kinase among Male Wrestlers"

_healthcare, 2023, doi:10.3390/healthcare11111643_

Round 1
Reviewer 1 Report
Dear Authors,
Your manuscript is interesting and certainly necessary.
I have a few comments.
In the introduction, please add a definition and determinants of depression. It cannot be indicated that it appears in players without its diagnosis (lines 55-53). Based only on the concentration of cortisol, it is impossible to ascertain. What about adrenal exhaustion and overtraining? It should also be indicated in the introduction what effect a single effort has on cortisol and what training. This is typical knowledge in the field of biochemistry of physical effort.
Is it influenced by mindfulness or changes in the stimulation of brain areas? How does this relate to the activity of the hypothalamus? Line 65-72
What factors affect cortisol levels? How does diet and glycogen content affect it? How does the type and duration of physical exertion affect it? This is not mentioned at all, and these are the main conditioning factors.
It is not stated in the introduction that CK can be strongly individual. It also depends on discipline, time, and the nature of physical effort.
Methodology
In the case of such a small group, it is recommended to carry out the Bonferroni correction.
The physical effort performed by the athletes during the study was not described. Its duration, intensity, training period, and these are very important issues. Diet and supplementation is also an important issue. It was also not given what the subjects did 72 hours before the test, which also has a significant impact on hormone levels.
Discussion
Other factors that may affect cortisol and CK levels in the body are not discussed at all.
Other factors that may affect cortisol and CK variability should also be considered. Exercise, training, diet, other hormone concentrations, etc.
The baseline level of these indicators should also be assessed and related to the baseline values of the population.
What are the reasons for the statistically significant changes, and where have they not occurred? How can you comment on that?
Conclusion
Conclusions must be based strictly on irrefutable evidence. In my opinion, the study protocol or the current discussion does not make it possible.
Kind regards,
The Reviewer

Author Response
We thank Reviewer #1 for their valuable and encouraging comments, which helped us to improve the quality of the manuscript. Please find the revised manuscript and the detailed point-by-point-response attached as separate files. Again, we thank Reviewer #1 for the care devoted to the present manuscript.

Reviewer 2 Report
Many thanks to the editors of the journal Healthcare for giving me the opportunity to review the article entitled: The Effect of A Standardized Mindfulness-Based Stress Reduc-2 tion (MBSR) Program on Symptoms of Stress and Depression 3 and on Saliva Cortisol and Serum Creatine Kinase among Male 4 Wrestlers
First of all, congratulations on the research work carried out, I would like to highlight the originality of the research question. I am also aware of how difficult it is to carry out research without funding support. I will now mention a number of recommendations in order to obtain clearer and more accurate information on your results.
1. In the introduction section, The introduction is quite complete but I would recommend not separating the paragraph on objectives and hypotheses from the rest of the introduction.
2. In the material and methods section, sub-section participants. It is not clear whether there are dropouts during the intervention and a flowchart illustrating the N of each phase and clearly indicating the reasons for exclusion and inclusion in the study should be included.
3. On the other hand, the power ad hod samples should be included so that we can be more precise in our conclusions or else they should be more cautious. I recommend the G-Power.(https://www.psychologie.hhu.de/arbeitsgruppen/allgemeine-psychologie-und-arbeitspsychologie/gpower)
4. Finally, in this section I recommend the use of the TIDieR-Checklist to ensure that certain incomplete aspects of the protocol description are covered, which make it difficult to translate. This checklist can be attached as supplementary material.
5. In the Analytical plan section, you must include a bibliographic reference of the effect sizes.
6. In the results section, I believe that the graphics can be scaled down and merged into a single image. In the image caption indicate what each one corresponds to in this way A) B) C) D).
7. In the discussion section, I encourage the inclusion of more references that allow for a more in-depth comparison and discussion of the data obtained.
8. The conclusion, I recommend separating the conclusions from the discussion and making a section of your own.
9. In the conclusions you state that it is an effective method for wrestlers, in order to make that conclusion you must first analyse the power of your sample, that way you will know if your study sample is representative of the study population. So please again incorporate a G-POWER ad hod analysis. And depending on the result modify the conclusions.
I hope these observations will be of help to you in your original study.
With best regards.
Author Response
We thank Reviewer #2 for their valuable and encouraging comments, which helped us to improve the quality of the manuscript. Please find the revised manuscript and the detailed point-by-point-response attached as separate files. Again, we thank Reviewer #1 for the care devoted to the present manuscript.

Round 2
Reviewer 1 Report
Dear Authors,
You have introduced additional information in the text which deserves attention. However, the manuscript, research methodology, and description still do not allow for an objective assessment of changes. The key is a thorough analysis and description of the basics of the study, such as diagnosing depression. You can't rely on the opinion of the participants whether they have it or not. Adding a note to the limitations will not improve the reliability of this manuscript. The second component, i.e. training, has also not been accurately described. Re-inclusion of the study limitation will not affect its validity. The study, although interesting, is not properly developed. You should add at least training summaries (volume, intensity of effort, you, character), describe the training period, indicate an individual’s diet, and assess your mental state. Without these descriptions, the results are unreliable and easy to challenge. Therefore, they do not contribute anything to science and do not enrich it.
Kind regards,
Reviewer
Author Response
We thank Reviewer #1 once again for their valuable and encouraging comments, which helped us to improve the quality of the manuscript. Please find the revised manuscript and the detailed point-by-point-response attached as separate files. Again, we thank Reviewer #1 for the care devoted to the present manuscript.

Reviewer 2 Report
Congratulations for the changes made, this makes the manuscript gain in methodological quality. But in the conclusion point, deleting this part is not the right solution. He encourages you to formulate this section apart from the discussion as a "5. Conclusions" section. And reword them along the lines of the conclusions section of the abstract.
After this simple correction we have no further objection to its publication.
Author Response
Again, we thank Reviewer #2 for their valuable and encouraging comments, which helped us to improve the quality of the manuscript. Please find the revised manuscript and the detailed point-by-point-response attached as separate files. Again, we thank Reviewer #1 for the care devoted to the present manuscript.
